# The Study of pH Effects on Phase Transition of Multi-Stimuli Responsive P(NiPAAm-*co*-AAc) Hydrogel Using 2D-COS

**DOI:** 10.3390/polym13091447

**Published:** 2021-04-29

**Authors:** Yeonju Park, Minkyoung Kim, Hae-jin Chung, Ah-hyun Woo, Isao Noda, Young-mee Jung

**Affiliations:** 1Kangwon Radiation Convergence Research Support Center, Kangwon National University, Chuncheon 24341, Korea; yeonju4453@kangwon.ac.kr; 2Department of Chemistry, Institute for Molecular Science and Fusion Technology, Kangwon National University, Chuncheon 24341, Korea; alsrud9351@naver.com (M.K.); jhyejin1212@gmail.com (H.-j.C.); dndkgus99@naver.com (A.-h.W.); 3Department of Materials Science and Engineering, University of Delaware, Newark, DE 19716, USA; noda@udel.edu

**Keywords:** multi-stimuli responsive polymer, phase transition, 2D-COS, 2D gradient mapping, PCA

## Abstract

The temperature and mechanism of phase transition of poly(*N*-isopropylacrylamide-*co*-acrylic acid) [P(NiPAAm-*co*-AAc)], which is one of the multi-stimuli responsive polymers, were investigated at various pHs using infrared (IR) spectroscopy, two-dimensional (2D) gradient mapping, and two-dimensional correlation spectroscopy (2D-COS). The determined phase transition temperature of P(NiPAAm-*co*-AAc) at pH 4, 3, and 2 based on 2D gradient mapping and principal component analysis (PCA) showed that it decreases with decreasing pH, because COOH group in AAc changes with variation of pH. The results of 2D-COS analysis indicated that the phase transition mechanism of P(NiPAAm-*co*-AAc) hydrogel at pH4 is different from that at pH2 due to the effect of COOH group of AAc.

## 1. Introduction

Smart polymers are very attractive materials in the field of biological applications because they show sharp phase transition changes with various stimuli such as pH, temperature, light, humidity, and so on [1,2,3,4,5]. However, single-stimuli responsive polymers are not always easy to apply in the body where various conditions exist, such as temperature, pH, ionic concentration, electricity, and so on [6,7]. Thus, a multi-stimuli responsive polymer becomes an attractive polymer to accomplish the desired result in a complex physiological microenvironment [6,7]. It has broad applications such as biomimetics, drug delivery, soft actuators and robots, functional materials for 4D printing, etc. [8,9,10,11,12,13,14,15,16]. The Poly(*N*-isopropylacrylamide) (PNiPAAm)-based polymer is one of the possible multi-stimuli responsive polymers. PNiPAAm is one of the more responsive polymers, which has the lower critical solution temperature (LCST) near 32 °C in aqueous system [17,18,19,20]. Therefore, the PNiPAAm-based polymer basically undergoes a phase transition in respond to changes in an external temperature. PNiPAAm has been modified with a number of polymers for a variety of applications [7,8,9,10,11,12,13,14,21,22,23,24,25,26,27]. Shieh et al. [13] successfully synthesized a temperature-, pH-, and CO_2_-responsive PNiPAAm-based random copolymer, which is copolymerized with acrylic acid (AAc). They reported that its LCST was increased with increasing pH value and decreased with increasing CO_2_ pressure. They used IR spectra only as evidence of synthesis. There is a lack of understanding at the molecular level of what causes LCST differences with pH changes. Nakajima et al. [14] fabricated and characterized multi-stimuli responsive hydrogel fibers of a P(NIPAM-AAc)-based mixed solution using a microfluidic spinning system. Ahiadu et al. [22] investigated the temperature- and pH-response kinetic of PNiPAAm-based microgels with various acids, such as acrylic acid (AAc), ethylacrylic acid (EAAc), methacrylic acid (MAAc), and butylacrylic acid (BAAc). However, they have seldom studied an understanding of the phase transition mechanism and relationship between moiety variation and variable conditions at a molecular level. 

Two-dimensional correlation spectroscopy (2D-COS) [28,29] provides the detailed information of inter- or intramolecular interaction, correlation between the materials’ properties and perturbations, and determination of the sequential order of the events of the system. It is very helpful to understand the relationship between structural changes of polymers and their properties. To explore the thermal phase transition mechanism of PNiPAAm-based copolymers, Sun et al. [30] reported the thermal volume phase transition of the synthesized P(NiPAAm-*co*-AAc) hydrogel dissolved in D_2_O using IR spectroscopy, perturbation-correlation moving-window (PCMW) and 2D-COS. The phase transition temperature and mechanism of P(NiPAAm-*co*-AAc) hydrogel during the heating and cooling processes were clearly determined using 2D-COS, which is not catchable information only using the traditional spectroscopy. Park et al. [31] also investigated the thermal transition mechanism of synthesized P(NiPAAm-*co*-AAc) hydrogel using IR spectroscopy, principal component analysis (PCA) and 2D-COS. However, the transition mechanism and the correlation between the properties of P(NiPAAm-*co*-AAc) hydrogel and variations of pH are not fully understood yet. 

In this study, to explore the thermal phase transition of P(NiPAAm-*co*-AAc) hydrogels with different pHs, temperature-dependent IR spectra of P(NiPAAm-*co*-AAc) hydrogels dissolved in various pH aqueous solutions are measured. To determine their transition temperature and to clearly investigate the phase transition mechanism, two-dimensional (2D) gradient mapping, PCA, and 2D-COS were performed. 

## 2. Materials and Methods

P(NiPAAm-*co*-AAc) polymer 15 mol% content of AAc and 1 M hydrogen chloride (HCl) solution were purchased from Sigma-Aldrich Co Ltd. (St. Louis, MO, USA). P(NiPAAm-*co*-AAc) hydrogel was prepared dissolving in double-distilled water with 3 wt% concentration and then kept at room temperature (25 °C) overnight for its stability. The initial pH of P(NiPAAm-*co*-AAc) hydrogel was 4.14. The pH of P(NiPAAm-*co*-AAc) hydrogel was adjusted to pH 3 and 2 using a few amounts of HCl solution (pH = 0.05, within 10 μL). To measure the temperature-dependent IR spectra of P(NiPAAm-*co*-AAc) hydrogel at different pHs (4, 3, and 2), Nicolet^TM^ iS50 Fourier-transform infrared (FTIR) spectrometer (Thermo Fisher Scientific, Waltham, MA USA) was used with the heated transmittance accessory (CaF_2_ window, PIKE Technologies, Waltham, MA USA) and the deuterated triglycine sulfate (DTGS) detector. All IR spectra were collected during the heating and cooling processes at temperature ranges from 27–40 °C (interval 1 °C) with a 4 cm^−1^ spectral resolution and 1024 scans. All IR spectra were baseline-corrected and intensity-normalized before carrying out the 2D gradient mapping, PCA and 2D-COS. Baseline correction, intensity normalization, and PCA were performed using Solo 8.9.1 (Eigenvector Research, Inc., Washington, DC, USA) with MATLAB R2019b software (The Mathworks Inc., Natick, MA, USA). 2D gradient map and 2D correlation spectra were obtained using home-made code in MATLAB R2019b. In the synchronous and asynchronous 2D correlation spectra, positive and negative cross peaks, respectively, are indicated by red and blue lines. 

Synchronous (**Φ**) and asynchronous (**Ψ**) 2D correlation spectra can be expressed by matrix form. A spectral data set obtained systemically under an external perturbation (e.g., pH, temperature, time, pressure, etc.). The dynamic spectra (**A**) are represented by subtracting the reference spectrum from a spectral data set. The simple cross-correlation analysis applied to the dynamic spectra to obtained 2D correlation spectra is as follows.
**Φ** = **A**^T^**A**(1)
**Ψ** = **A**^T^**NA**(2)

Here, the element of the Hilbert–Noda transformation matrix N at i^th^ row and j^th^ column is given by 1/π (j − i), and zero if i = j. The synchronous 2D correlation spectrum shows the similarity between the spectral changes under external perturbation, and the corresponding asynchronous 2D correlation spectrum shows dissimilarity. The autopeak at the diagonal line of the synchronous 2D correlation spectrum indicates the overall spectral changes with perturbation changes, and the cross peak at the off diagonal line of the synchronous 2D correlation spectrum indicates the direction of the spectral intensity changes between correlation peaks. In the asynchronous 2D correlation spectra, only cross peaks appeared. By comparing the sign of the cross peaks at synchronous and asynchronous 2D correlation spectra, the sequential order of the spectral intensity changes can be determined. The detailed background of 2D-COS is referred to Noda et al. [28].

## 3. Results and Discussion

### 3.1. Analysis of the Temperature-Dependent IR Spectra of P(NiPAAm-co-AAc) Hydrogels with Different pHs

Temperature-dependent IR spectra of P(NiPAAm-*co*-AAc) hydrogels with different pHs (4, 3, and 2) were displayed in Figure 1. In Figure 1, a strong band at 1640 cm^−1^ is observed, corresponding to the water band. To eliminate the interference of this band, all spectra were intensity-normalized using intensity of this band. 

As shown in Figure 1a–d, the spectral changes of P(NiPAAm-*co*-AAc) hydrogels at pH 4 and 3 are quite similar. In the C-H stretching region (3010–2850 cm^−1^), the intensity of a band at 2982 cm^−1^ increases and those at 2933, 2917, and 2882 cm^−1^ decrease with increasing temperature (see Appendix A). Four bands at 2982, 2933, 2917, and 2882 cm^−1^, respectively, are assigned to the hydrated ν_as_(CH_3_), hydrated ν_as_(CH_2_), dehydrated ν_as_(CH_2_), and dehydrated ν_s_(CH_3_) [30,31]. Two bands at 2982 and 2882 cm^−1^ shift to lower wavenumbers (2977 and 2877 cm^−1^) with increasing temperature. During the cooling process, the trends in intensity variation and position shift of these bands are different compared to the heating process. In the 1800–1480 cm^−1^ spectral region, intensities of the two shoulder bands near 1688 and 1610 cm^−1^ and a band at 1566 cm^−1^ decrease with increasing temperature (see Appendix A). Three bands at 1688, 1610, and 1566 cm^−1^, respectively, were assigned to the intermolecular hydrogen bond of ν(C=O) in AAc (C=OAAc⸱⸱⸱H_2_-O), the intermolecular hydrogen bond of ν(C=O) in NiPAAm, (C=O_NiPAAm_⸱⸱⸱H_2_O), and hydrated δ(N-H) in NiPAAm (N-H⸱⸱⸱OH_2_). This observation shows evidence of the dehydrated P(NiPAAm-*co*-AAc) hydrogel formation during the heating process. The trend of band intensity changes during the cooling process (See Appendix A) is the opposite of that of the heating process. The overall trend of the spectral changes of P(NiPAAm-*co*-AAc) hydrogels at pH4, 3, and 2 is similar to each other. However, in the C-H stretching region, the trend of the spectral changes of P(NiPAAm-*co*-AAc) hydrogel at pH2 (see Figure 1e,f and Appendix A) is different from those at pH4 and 3. To more deeply explore the temperature-dependent IR spectra of P(NiPAAm-*co*-AAc) hydrogels with different pHs, 2D gradient mapping, PCA, and 2D-COS were performed.

### 3.2. Determination of Transition Temperature of P(NiPAAm-co-AAc) Hydrogel with Different pHs

To determine the transition temperature of P(NiPAAm-*co*-AAc) hydrogel at different pHs (4, 3, and 2), 2D gradient mapping analysis was carried out (Figure 2). 2D gradient mapping proposed by Jung et al. [32] is a very simple and intuitive method to detect the transition temperature of a polymer from its spectra. In the 2D gradient mapping method, the value of the first derivatives of absorbance with respect to temperature is represented on a 2D map that directly reflects the transition temperature at the location of the minima or maxima. As shown in Figure 2a,b, the transition temperatures of P(NiPAAm-*co*-AAc) hydrogel at pH4 are 33.5 and 35.5 °C during the heating and cooling processes, respectively. At pH3 and 2, that is 32.5 °C during both heating and cooling processes (see Figure 2c–f). We can also confirm the transition temperature from the score plots for the first two principal components (PCs) of the temperature-dependent IR spectra of P(NiPAAm-*co*-AAc) hydrogel with different pHs (4, 3, and 2) during the heating and cooling processes. Based on the scores of PC1, spectral changes of all bands occur between 33 and 34 °C for pH4, and between 32 and 33 °C for pH 3 and 2 during the heating process. During the cooling process, spectral changes of all bands are observed between 36 and 35 °C for pH4, between 33 and 32 °C for pH 3, and between 33 and 31 °C for pH 2. The determined transition temperature of P(NiPAAm-*co*-AAc) hydrogel decreases with decreasing pH. These temperatures determined by both 2D gradient mapping and scores are consistent with the study reported by Shieh et al. [13]. They noted that the COOH groups in AAc segments dissociated to COO^−^ units at higher pH, resulting in good solubility of P(NiPAAm-*co*-AAc) hydrogel in aqueous solutions. For this reason, the transition temperature decreases with decreasing pH.

### 3.3. The Phase Transition Mechanism of P(NiPiPAAm-co-AAc) Hydrogels with Different pHs during the Heating and Cooling Processes

To understand the phase transition mechanism of P(NiPAAm-*co*-AAc) hydrogel with different pHs (4 and 2) during the heating and cooling processes, 2D-COS was applied to its temperature-dependent IR spectra. Figure 3 and Figure 4 show the 2D correlation spectra at pH4 during the heating and cooling processes, respectively. In synchronous 2D correlation spectrum shown in Figure 3a, there are two autopeaks at 2971 and 2891 cm^−1^ in the C-H stretching region. This means that dehydrated ν_as_(CH_3_) and ν(CH) are the most sensitive components in P(NiPAAm-*co*-AAc) hydrogel at pH4 during the heating process. Four negative cross peaks at (2991,2971), (2945,2971), (2908,2971), (2867,2971) cm^−1^ are observed. It means that intensity changes of a band at 2971 cm^−1^ is opposite to those of four bands at 2991, 2945, 2908, and 2867 cm^−1^. 2D-COS provides the useful information of the sequential order of the spectral changes under an external perturbation [28]. If the signs of the cross peaks in synchronous and asynchronous 2D correlation spectra are the same, the spectral intensity changes at ν_1_ occurs before that at ν_2_. While, if they are different, the spectral intensity changes at ν_1_ occurs after that at ν_2_. As shown in the Figure 3a, the signs of cross peaks at (2887, 2976) cm^−1^ in the synchronous and asynchronous 2D correlation spectra are the same. Therefore, the intensity change of the band at 2887 cm^−1^ occurs before that of the band at 2976 cm^−1^. Based on the 2D correlation analysis, the sequential order of the intensity changes of corresponding bands with increasing temperature was determined: 2870 → 2887 → 2946 → 2976 → 2992 cm^−1^. This result indicates that the intensity of the dehydrated ν_s_(CH_3_) group firstly changes, and then intensity variation of the hydrated ν_s_(CH_3_) group changes before that of the hydrated ν_as_(CH_2_) group. Finally, the intensity of ν(CH) group changes. Figure 3b displays 2D correlation spectra in the C=O stretching region coupled with N-H bending region. In the synchronous 2D correlation spectra, there are two positive cross peaks at (1610, 1682) and (1576, 1682) cm^−1^, indicating that the intensities of three bands at 1682, 1610, and 1576 cm^−1^ decrease together during the heating process. In the asynchronous 2D correlation spectra, new bands at 1720, 1631, and 1543 cm^−1^ are observed, which are hardly detected in 1D spectra (see Figure 1a). They are assigned to intramolecular interaction between AAc and NiPAAm (C=O_AAc_⸱⸱⸱H-N_NiPAAm_), intramolecular hydrogen bond of ν(C=O) in NiPAAm (C=O⸱⸱⸱H-N), dehydrated δ(N-H) in NiPAAm (N-H⸱⸱⸱O=C), respectively. From the 2D correlation analysis, we determine the sequence of the band intensity changes with increasing temperature: 1720 → 1682 → 1610 → 1566 → 1631 → 1539 cm^−1^. The intensity changes of AAc related components occurs before those of NiPAAm related components. This means that the AAc can be delayed the phase transition of PNiPAAm during heating process. The overall sequential order of the intensity changes of P(NiPAAm-*co*-AAc) hydrogel at pH4 during the heating process is determined using hetero-mode 2D correlation spectra (see Figure 3c): 1720 → 1682 → 1610 → 1566 → 1631 → 1539 → 2870 → 2887 → 2946 → 2976 → 2992 cm^−1^. This result means that the components corresponding to C=O stretching coupled with N-H bending modes in P(NiPAAm-*co*-AAc) hydrogel changes before that to C-H stretching modes during the heating process.

Figure 4 illustrates the corresponding 2D correlation spectra during the cooling process. As shown in the Figure 4a, two autopeaks at 2989 and 2972 cm^−1^ are observed in synchronous 2D correlation spectrum. Unlike the heating process, hydrated ν_as_(CH_3_) and dehydrated ν_as_(CH_3_) are more sensitive than the other C-H stretching modes during the cooling process. This indicates that the specific group, which changes most during the heating process, is different to that during the cooling process. From the analysis of 2D correlation spectra, the sequence of the spectral intensity changes during the cooling process can be also determined as follows: 2986 → 2942 → 2887 → 2973 cm^−1^. From the analysis of 2D correlation spectra for the C=O stretching coupled with N-H bending modes (1800–1480 cm^−1^) shown in Figure 4b, the sequential order of spectral intensity changes during the cooling process is similar with that during the heating process sequence: 1613 → 1686 → 1565 → 1636 → 1543 cm^−1^. Moiety of AAc changes before that of NiPAAm. From the analysis of the hetero-mode 2D correlation spectra (see Figure 4c), we can explain the full scenario of the intensity changes of P(NiPAAm-*co*-AAc) hydrogel during the cooling process. The moiety of AAc changes firstly, then the moiety of NiPAAm changes, and finally the C-H stretching region changes with decreasing temperature. From the results of 2D-COS, we can assume that the change in AAc played a role in driving force to change the phase transition of PNiPAAm hydrogel during the heating and cooling processes.

Figure 5 and Figure 6 represent the 2D correlation spectra of P(NiPAAm-*co*-AAc) hydrogel at pH2 during the heating and cooling processes. Two autopeaks at 2972 and 2932 cm^−1^ related with dehydrated ν_as_(CH_3_) and dehydrated ν_as_(CH_2_) are prominently observed in synchronous 2D correlation spectra during the heating and cooling process (see Figure 5a and Figure 6a). This observation is different with Figure 3a and Figure 4a. The sequential order of the intensity changes in the C-H stretching region during the heating process are opposite to those during the cooling process: 2981 → 2935 → 2907 → 2968 cm^−1^ during the heating process; 2961 → 2900 → 2925 → 2935 → 2877 → 2915 → 2978 cm^−1^ during the cooling process. As shown in Figure 5b and Figure 6b, the results of the 2D correlation analysis in C=O stretching coupled with N-H bending region are also different from pH4. In the synchronous 2D correlation spectra during the heating and cooling processes, a cross peak at (1688,1758) cm^−1^ is observed. This indicates that the portion of intramolecular hydrogen bond of ν(C=O) between AAc and PNiPAAm (C=O_AAc_⸱⸱⸱H-N_NiPAAm_) of P(NiPAAm-*co*-AAc) hydrogel at pH2 is much more than that at pH4. Unlike the pH4, the phase transition mechanism during the heating and cooling processes shows opposite each other: 1617 → 1679 → 1539 → 1735 cm^−1^ during the heating process; 1708 → 1563 → 1680 → 1630 → 1546 cm^−1^ during the cooling process. A new band at 1708 cm^−1^ appears at pH2, which is assigned to the intermolecular hydrogen bond of AAc (C=O_AAc_⸱⸱⸱H-O_AAc_). Based on the hetero-mode 2D correlation spectra (see Figure 5c,d), we concluded the full phase transition mechanism of P(NiPAAm-*co*-AAc) hydrogel at pH2 during the heating and cooling processes as follows: intermolecular hydrogen bond between H_2_O and AAc (or NiPAAm) converts to an intramolecular hydrogen bond, and then dehydration of the C-H stretching region changes, and finally the intramolecular interaction between AAc and NiPAAm is broken during the heating process; the formation of the intermolecular hydrogen bond of AAc (C=O_AAc_⸱⸱⸱H-O_AAc_) is before that of hydrated δ(N-H) (N-H⸱⸱⸱O-H_2_), intermolecular hydrogen bond between AAc and H_2_O increases, then C-H stretching region changes, and lastly the intramolecular hydrogen bond of NiPAAm (C=O⸱⸱⸱H-N) is broken during the cooling process.

From the results of 2D correlation spectra, we successfully determined the thermal phase transition mechanism of P(NiPAAm-*co*-AAc) hydrogels at different pHs. These results show that the phase transition of P(NiPAAm-*co*-AAc) hydrogel is irreversible process at various pHs. The differences of two hydrogels also can be well explained, although their 1D spectra look similar. The introduction of AAc to the PNiPAAm makes a different phase transition mechanism of PNiPAAm hydrogel as pH changes. In other words, AAc greatly contributes to the changes in the chemical properties of PNiPAAm hydrogel. Therefore, P(NiPAAm-*co*-AAc) hydrogel is applicable as the multi-stimuli responsive polymer.

## 4. Conclusions

In this study, the transition temperature of P(NiPAAm-*co*-AAc) hydrogels at pH4, 3, and 2 is successfully determined using 2D gradient mapping and PCA. Their transition temperature is decreased with decreasing pH, resulting from dissociation of COOH in AAc at higher pH. From 2D correlation analysis, the sequential order of band intensity changes of P(NiPAAm-*co*-AAc) hydrogels at pH4 and 2 is successfully determined. Therefore, we can conclude that the phase transition of P(NiPAAm-*co*-AAc) hydrogels take place irreversibly, even if they show the different thermal phase transition mechanism. The results of 2D-COS and 2D gradient mapping provide the new insight of understanding the phase transition mechanism of multi-stimuli responsive polymers, high functionality polymers, and various types of polymers.

## Figures and Tables

**Figure 1 polymers-13-01447-f001:**
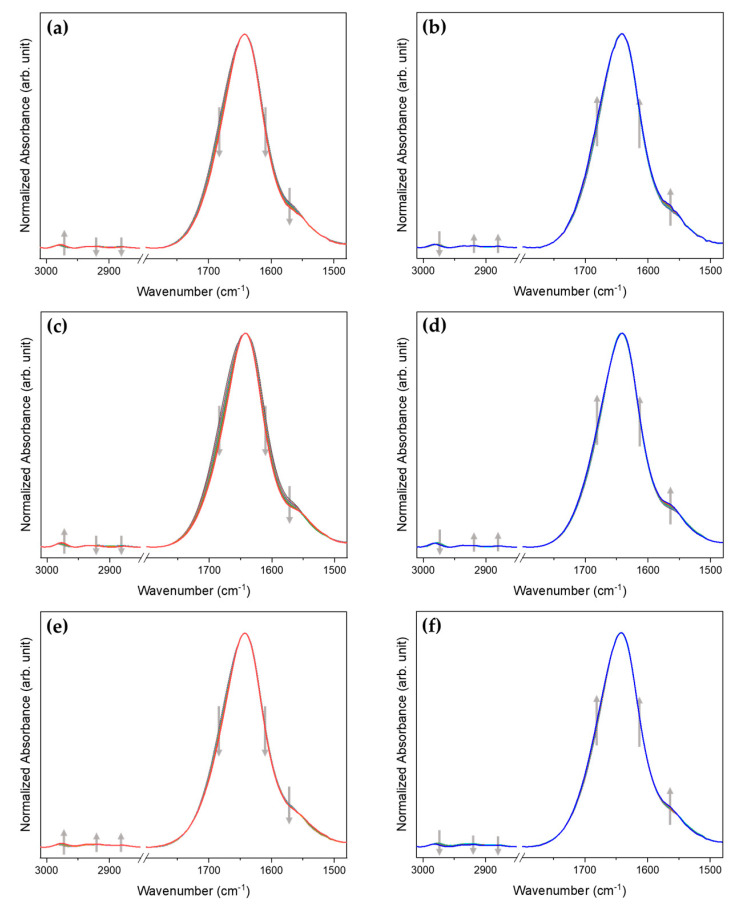
Temperature-dependent IR spectra of P(NiPAAm-*co*-AAc) hydrogel of pH4 (**a**,**b**), pH3 (**c**,**d**), pH2 (**e**,**f**) during the heating (**a**,**c**,**e**) and cooling (**b**,**d**,**f**) processes. The gray arrows indicate the direction of intensity change with varying temperature; from 27 to 40 °C is during the heating process and from 40 to 27 °C is during the cooling process.

**Figure 2 polymers-13-01447-f002:**
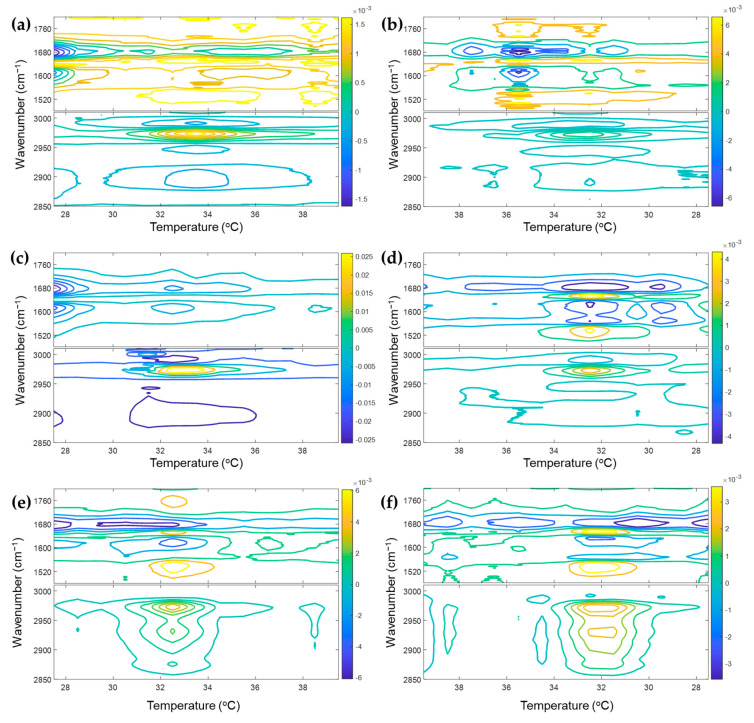
2D gradient maps obtained from temperature-dependent IR spectra of P(NiPAAm-*co*-AAc) hydrogel with different pHs (**a**,**b**: 4, **c**,**d**: 3, **e**,**f**: 2). Heating processes are (**a**,**c**,**e**), and the cooling process are (**b**,**d**,**f**).

**Figure 3 polymers-13-01447-f003:**
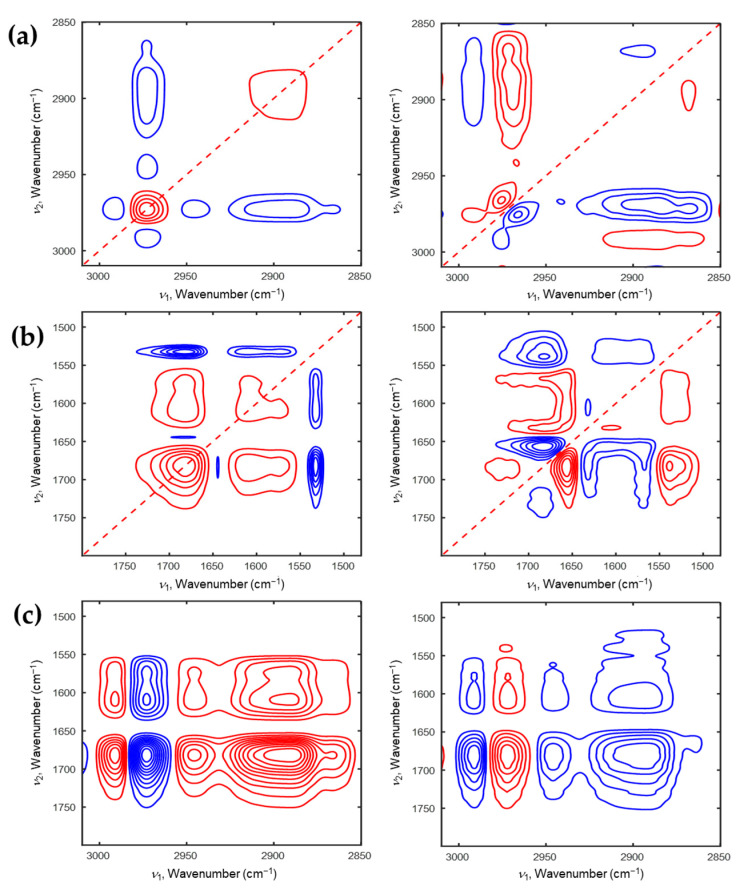
2D correlation spectra of P(NiPAAm-*co*-AAc) hydrogel of pH4 during the heating process (**a**: 3010–2850 cm^−1^, **b**: 1800–1480 cm^−1^). Hetero-mode 2D correlation spectra (**c**) of P(NiPAAm-*co*-AAc) hydrogel during the heating process for the 3010–2850 cm^−1^ and 1800–1490 cm^−1^ regions.

**Figure 4 polymers-13-01447-f004:**
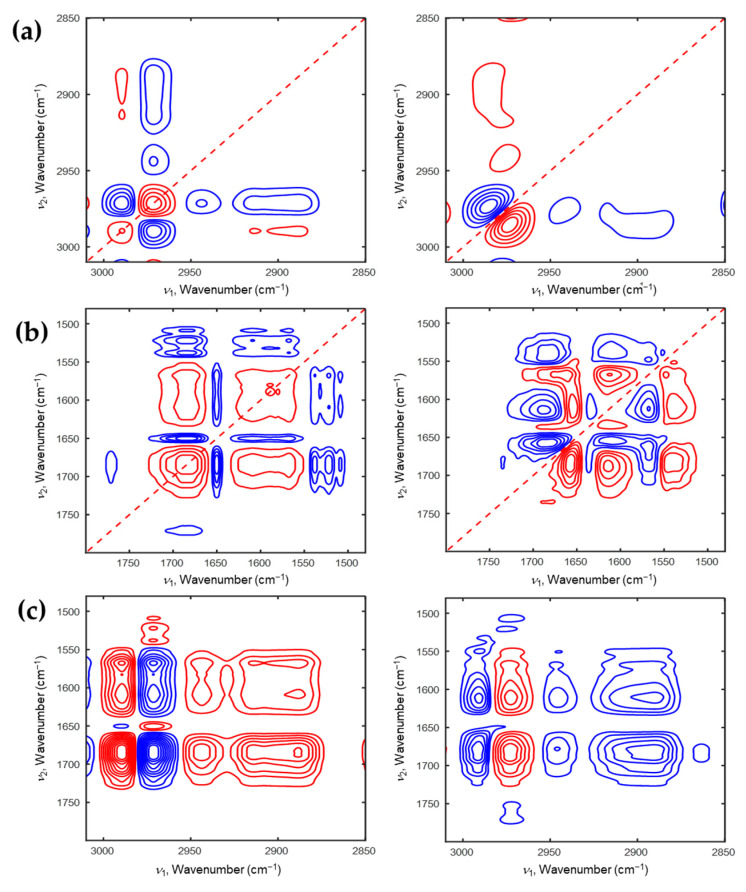
2D correlation spectra of P(NiPAAm-*co*-AAc) hydrogel of pH4 during the cooling process (**a**: 3010–2850 cm^−1^, **b**: 1800–1480 cm^−1^). Hetero-mode 2D correlation spectra (**c**) of P(NiPAAm-*co*-AAc) hydrogel during the cooling process for the 3010–2850 cm^−1^ and 1800–1490 cm^−1^ regions.

**Figure 5 polymers-13-01447-f005:**
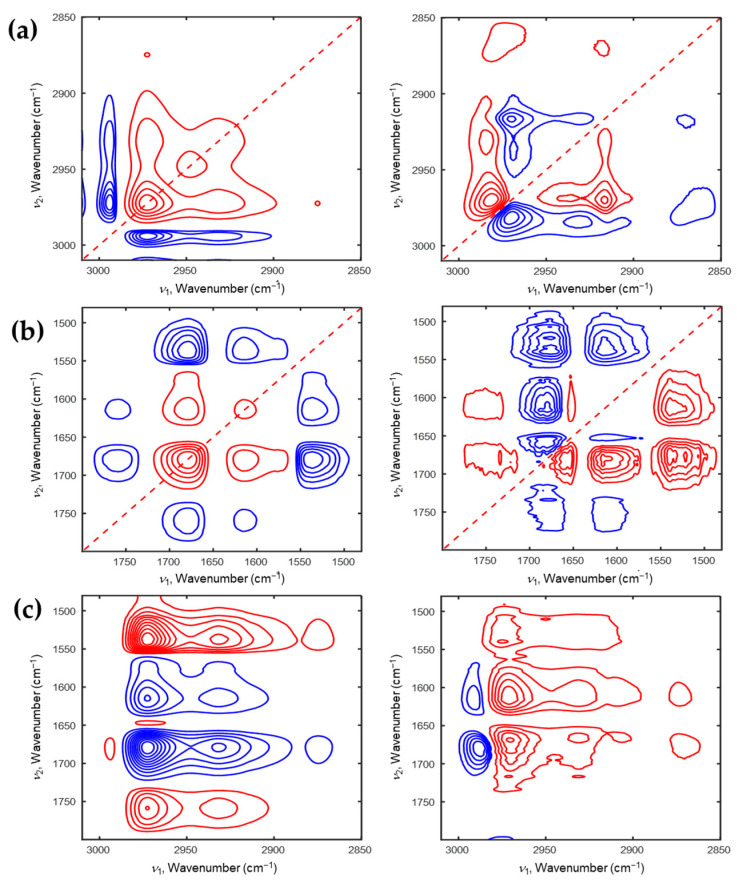
2D correlation spectra of P(NiPAAm-*co*-AAc) hydrogel of pH2 during the heating process (**a**: 3010–2850 cm^−1^, **b**: 1800–1480 cm^−1^). Hetero-mode 2D correlation spectra (**c**) of P(NiPAAm-*co*-AAc) hydrogel during the heating process for the 3010–2850 cm^−1^ and 1800–1490 cm^−1^ regions.

**Figure 6 polymers-13-01447-f006:**
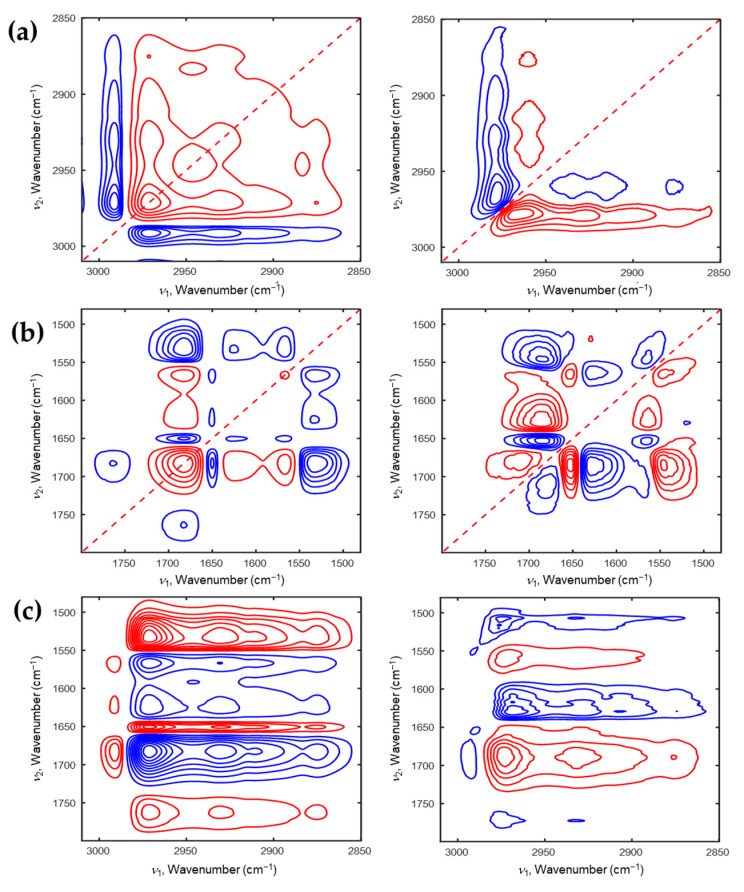
2D correlation spectra of P(NiPAAm-*co*-AAc) hydrogel of pH2 during the cooling process (**a**: 3010–2850 cm^−1^, **b**: 1800–1480 cm^−1^). Hetero-mode 2D correlation spectra (**c**) of P(NiPAAm-*co*-AAc) hydrogel during the cooling process for the 3010–2850 cm^−1^ and 1800–1490 cm^−1^ regions.

## Data Availability

The data presented in this study are available on request from the corresponding author.

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
