# Peer review of "The Study of pH Effects on Phase Transition of Multi-Stimuli Responsive P(NiPAAm-*co*-AAc) Hydrogel Using 2D-COS"

_polymers, 2021, doi:10.3390/polym13091447_

Round 1
Reviewer 1 Report
This study used 2DCOS and PCA determined the transition temperature of P(NiPAAm-co-AAc) hydrogels at differetn pH values. The novelty, significance and quality of this manuscript are fine. However, the language can be improved for better understanding.
Author Response
The authors appreciate the positive comment. According to Reviewer’s comments, we have improved the text under the help of a native English speaker.
Reviewer 2 Report
The authors have dealt with a system of considerable current interest, i.e. multi-stimuli responsive polymer. These materials in this paper consist P(NiPAAm-co-AAc) with different contents, that are LCST phase transition via kinds of pH-dependent and temperature-dependent. Hence, the thermodynamics of LCST phase transition and experiments play a crucial role in deciding the properties of the copolymers. Meanwhile, it is very commendable that they have carried out detailed experiments (To determine their transition temperature and to clearly investigate phase transition mechanism, two-dimensional (2D) gradient mapping, PCA, and 2D-COS were performed.) and laid a solid foundation for theoretical analysis. However, some questions should be addressed. The detailed comments are listed as follows;
1. For polymers-oriented journals, the reader expects to see a more focused introduction, with a short description of the state-of-the-art of multi-stimuli responsive polymer, a detailed formulation of the problem, and methods used to tackle them. I do not see this in the current manuscript.
2. In Figure 1, the curve of cooling can be changed to blue to distinguish heating, and the curve should be bold due to the Figure is not clear.
3. The 2D-COS analysis provided in this paper has certain significance for LCST phase transition, we suggest that the 2D-COS analysis should be introduced in “2. Materials and Methods”.
4. The phase transition mechanism in this paper is too weak, since it is suggested to refer to the rich contents. The following literature is for reference only. (https://doi.org/10.1007/s10409-021-01072-4 & https://doi.org/10.1088/1361-665X/ab9e0c )
5. The paper seriously lacks in clarity. The most important features of the proposed model are not at all clear. This should be brought out at this level, i.e., in the very beginning.
Author Response
All the replies are written in boldface.
We appreciate Reviewers’ comments on our manuscript which have made us revise the manuscript significantly.
Our point-by-point response to the comments of Reviewer is included.

Reviewer 3 Report
I have reviewed a manuscript “The study of pH effects on phase transition of P(NiPAAm-co-AAc) hydrogel using 2D-COS”. The work aimed to explore the thermal phase transition of P(NiPAAm-co-AAc) hydrogels with different pHs. This work provides very interesting results with adequate results. I think it is suitable for publication after addressing the following comments:
Comment 1: Is 2D-COS method is applicable for other thermo-responsive hydrogels?
Comment 2: Please extend the introduction and discuss more the application of thermo-responsive hydrogels such as NIPAM, Pluronic, etc. following references might be helpful:
- https://www.sciencedirect.com/science/article/pii/S0939641107002470
- https://www.sciencedirect.com/science/article/pii/S0939641115002490
- https://onlinelibrary.wiley.com/doi/full/10.1002/pat.4654
- https://www.ncbi.nlm.nih.gov/pmc/articles/PMC6415431/
- https://pubmed.ncbi.nlm.nih.gov/17881200/
Comment 3: Please highlight the potential application of 2D-COS method in the characterization of thermo-responsive hydrogel in the introduction in detail.
Author Response
All the replies are written in boldface.
We appreciate Reviewers’ comments on our manuscript which have made us revise the manuscript significantly.
Our point-by-point response to the comments of Reviewers is included.

Round 2
Reviewer 3 Report
The authors have addressed the comments very well. It is suitable for publication as it is.
Author Response
We appreciate very much for the positive comment.